# Client satisfaction with pharmacy services and associated factors at Yekatit 12 Hospital Medical College, Addis Ababa, Ethiopia

**Mekdes Tamiru Yizengaw**[1]*, **Bekele Simegn Demissie**[1], **Beimnet Taye Mekonnen**[1], **Birknesh Tamire Haile**[1], **Sosina Fikadu Mulugeta**[1], **Ekram Kemale Mohmmed**[1], **Samrawit Birhanu Alemu**[2], **Melaku Birhanu Alemu**[3,]

1 St. Lideta health science and business college, Addis Ababa, Ethiopia, 2 Department of Public Health, Debre Markos University, Debre Markos, Ethiopia, 3 Department of Health Systems and Policy, Institute of Public Health, University of Gondar, Gondar, Ethiopia

* mekdesmela@gmail.com

## Abstract

### Background

Client satisfaction is a crucial indicator of healthcare quality, influencing treatment adherence, continuity of care, and overall health outcomes. However, there is limited evidence on client satisfaction with pharmacy services in Ethiopia. This study aims to assess client satisfaction levels and their determinants in pharmacy services at Yekatit 12 Hospital, Addis Ababa.

### Methods

A hospital-based cross-sectional study was conducted at Yekatit 12 Hospital Medical College, from 21 April 2025–15 May 2025. A total of 422 adult patients and caregivers were selected using systematic random sampling from five pharmacy units. Data were collected through a structured, interviewer-administered questionnaire. The questionnaire included sociodemographic characteristics and measures of satisfaction. Responses were measured on a five-point Likert scale, and the mean score was used to categorise participants as satisfied or dissatisfied. Data were analysed using STATA software, and multivariable logistic regression identified factors associated with satisfaction. Ethical approval was obtained from the Institutional Review Board (IRB) of Yekatit 12 Hospital Medical College (Reference number: RPO/934/25).

### Results

Overall, 49.76% of participants were satisfied with outpatient pharmaceutical services. Participants with secondary education or higher had significantly lower odds of satisfaction compared with those with no formal education (AOR = 0.23; 95% CI: 0.07–0.79). Clients who did not receive all prescribed medications were also less

**Data availability statement:** All relevant data are within the paper and its Supporting Information files.

**Funding:** The author(s) received no specific funding for this work.

**Competing interests:** The authors have declared that no competing interests exist.

likely to be satisfied (AOR = 0.35; 95% CI: 0.20–0.60). In contrast, ART pharmacy users had higher odds of satisfaction compared with outpatient pharmacy users (AOR = 3.27; 95% CI: 1.38–7.78).

## Conclusion

Client satisfaction with pharmacy services at Yekatit 12 Hospital was low. Educational status, type of pharmacy service, and medication availability were associated with satisfaction levels. Higher satisfaction in the ART pharmacy underscores the value of specialised service. Strengthening drug supply systems, enhancing staff-client interaction, and improving service processes may increase satisfaction across pharmacy units.

---

## Introduction

Pharmacy services play a crucial role in the healthcare system, significantly contributing to public health and favourable health outcomes [1]. In recent years, pharmacy services have evolved from solely dispensing medications to adopting a more patient-centred approach focused on comprehensive care [2].

Healthcare quality has become a major global concern, evolving to meet patients' growing needs [3,4]. Poor quality treatment is responsible for around 5.7 to 8.4 million fatalities annually in low- and middle-income countries (LMICs) [5]. Sub-Saharan Africa disproportionately has a lower quality of health, with only 42.4% being satisfied with the availability of quality health care [6].

While it was traditionally assessed using professional practice standards, the inclusion of broader quality indicators has made patients' perceptions an increasingly important measure of healthcare performance [7]. Ensuring the accessibility of pharmacy services and maintaining client satisfaction are fundamental aspects of enhancing the overall quality of pharmaceutical care provided [8–10]. Despite the significant role of client satisfaction often being low in Ethiopia, with outpatient pharmacy service satisfaction ranging from 46.2% to 59.4% [1,11].

Clients who experience satisfaction with pharmaceutical services are more likely to adhere to their prescribed medications and less likely to seek care from different places [12]. Evidence from numerous studies indicates that higher levels of patient satisfaction with healthcare services are linked to improved adherence to prescribed treatments, enhanced health outcomes, and more efficient use of healthcare resources [13]. Client experiences in healthcare settings are diverse, and multidisciplinary collaboration among healthcare providers plays a critical role in enhancing both satisfaction and overall health outcomes [14].

Various studies found that Sociodemographic characteristics, including gender, age, occupation, and educational background, influence pharmacy service satisfaction [1,11,15,16]. Moreover, satisfaction was associated with drug availability, waiting time and the comfort of the waiting area [1,17]. Although several studies in Ethiopia have examined client satisfaction with general healthcare services [1,11,15], there is a notable scarcity of data specifically addressing client satisfaction with pharmacy services

across different regions of the country [18]. No recent study has been conducted at Yekatit 12 Hospital in Addis Ababa to assess client satisfaction levels and their determinants in pharmacy services. Therefore, this study aims to assess the magnitude of client satisfaction and the contributing factors in pharmacy services at Yekatit 12 Hospital in Addis Ababa, Ethiopia.

## Methods

### Study setting

This study was conducted at Yekatit 12 Hospital Medical College, a public referral hospital and one of the largest teaching hospitals in Addis Ababa, Ethiopia. The hospital provides a broad range of specialised healthcare services while also serving as a medical training centre. The hospital was founded in 1923. According to the hospital report, the hospital supports 342,787 community-based health insurance (CBHI) members and manages an annual prescription volume of nearly 446,000. The hospital operates with a bed capacity of 308 and employs 72 pharmacists who are actively involved in delivering pharmaceutical services. Each year, the hospital manages around 3,000 road traffic accident (RTA) cases and provides training for 562 medical students.

### Study design and period

A hospital-based cross-sectional study was conducted to assess client satisfaction with pharmacy services and associated factors at Yekatit 12 Hospital in Addis Ababa, Ethiopia. The study was conducted over a defined period, from 21 April 2025–15 May 2025.

### Source population

The source population for this study included all patients and caregivers who utilised pharmacy services at Yekatit 12 Hospital in Addis Ababa, Ethiopia. This encompassed individuals who visited various pharmacy service units, including the Outpatient Department (OPD) pharmacy, Antiretroviral Therapy (ART) pharmacy, Inpatient Department (IPD) pharmacy, paediatric pharmacy, and the emergency pharmacy, for medication dispensing, counselling, and other pharmaceutical care services. Caregivers who collected medications or received counselling on behalf of in-patients or paediatric patients were also included, as they represent an important group receiving pharmacy services within the hospital system.

### Study population

The study population consisted of adult clients aged 18 years and above who visited the pharmacy services at Yekatit 12 Hospital in Addis Ababa, Ethiopia, during the study period. These individuals included clients who interacted with any of the hospital's pharmacy service units, such as those seeking prescription dispensing, medication counselling, and other pharmaceutical care services. The study focused on clients who utilised pharmacy services during the specified timeframe.

### Inclusion and exclusion criteria

**Inclusion criteria:** Clients aged 18 years and above who use pharmacy services at Yekatit 12 Hospital during the data collection period.
**Exclusion criteria:** Clients who were severely ill and unable to respond to the questionnaire; Clients with hearing or speech impairments that prevented effective communication; Clients aged below 18 years.

### Sample size determination

The sample size for this study was determined using the single population proportion formula:

$$n = \frac{z^2 p(1-p)}{d^2}$$

The formula represents: $n$ = Sample size; Z = 1.96; p = 51.6%, taken from a recent study reported on patient satisfaction with outpatient pharmacy service at Black Lion Specialised Hospital [19], d = marginal error assumed to be 5%.

$$n = \frac{(1.96)^2 * 0.516 * (1 - 0.516)}{(0.05)^2}$$

The calculated minimum sample size needed for this study was 384 respondents. We assumed a 10% non-response rate (38 respondents), yielding a final sample of 422.

## Sampling technique and procedures

In this study, probability systematic random sampling was employed to select participants from all pharmacy units at Yekatit 12 Hospital. This method was chosen to ensure an unbiased and representative selection of patients, thereby enhancing the reliability of the study findings. The first step in this sampling technique involved determining the sampling interval (K), calculated using the formula K = N/n, where N represents the estimated total number of patients visiting the Outpatient Department (OPD) pharmacy, Antiretroviral Therapy (ART) pharmacy, Inpatient Department (IPD) pharmacy, paediatric pharmacy, and emergency pharmacy during the data collection period, and n is the final sample size, which was 422. Based on the hospital records, approximately 2,000 patients visit all pharmacy services daily. Using this figure, the sampling interval was calculated as K = 2000/422 ≈ 5, meaning that every 5th patient was selected to participate in the study.

To ensure randomness, the first participant was selected at random from the first K individuals using a lottery method. The lottery method selected the 3rd patient as the starting point; therefore, the 3rd patient in the sequence was chosen as the first participant, and thereafter every 5th patient was included until the target sample size was reached.

## Data collection instrument

Data was collected using a structured interview questionnaire, which was developed based on standard relevant literature on the subject [10, 11]. The questionnaire was divided into two main sections. The first section collected sociodemographic information from participants, including age, gender, place of residence, marital status, educational level, and occupation, to identify potential demographic factors that may influence patient satisfaction with pharmacy services (S1 File).

The second section of the questionnaire was organised under four thematic subheadings. The first subheading assessed patients' experiences with pharmacy services, including waiting time, professionalism of staff, and overall service satisfaction. The second subheading focused on the pharmacy setting, such as medication availability, prescription costs, and the physical environment of the pharmacy. The third subheading addressed pharmacist-patient communication, evaluating the clarity and quality of counselling provided by the pharmacists. The final subheading measured overall satisfaction with the pharmacy services during the most recent visit.

For all satisfaction-related items, a five-point Likert scale was used, where 1 = Very Satisfied, 2 = Satisfied, 3 = Neutral, 4 = Dissatisfied, and 5 = Very Dissatisfied. For analysis and descriptive interpretation, the responses were categorised into a three-level scale: "Very Satisfied" and "Satisfied" were grouped under "Satisfied", while "Dissatisfied" and "Very Dissatisfied" were grouped as "Dissatisfied". The "Neutral" category was left unchanged.

The questionnaire was initially prepared in English and then translated into Amharic to ensure clarity and comprehension among participants. To maintain content accuracy, the Amharic version was translated back into English. The tool was a standardised questionnaire based on previous literature, so pre-testing on a sample of participants was not performed. Data collection was conducted using the Kobo Toolbox online survey platform, which helped enhance data quality and facilitate organised entry and analysis.

 

## Study variables

**Dependent variable.** Patient satisfaction with pharmacy services, categorised as *satisfied* or *dissatisfied*. Participants with satisfaction scores above the mean score were classified as satisfied, while those with scores at or below the mean were classified as dissatisfied.

**Independent variables.** **Demographic variables**: Age, gender, occupation, education level, marital status, place of residence, wealth, region

**Pharmacy service variables**: Waiting Time, Pharmacy Environment, Medication Availability, Cost of Medications, Pharmacist's Professionalism, Pharmacist-Patient Communication

**Service delivery variables**: Medication Counselling, Accessibility of Pharmacy Services

## Data quality control

To ensure the accuracy and reliability of the data, several quality control measures were implemented throughout the study. During the data collection period, activities were conducted under close supervision to ensure strict adherence to the study protocol. Data was entered using a double-entry system to reduce input errors. Following data entry, a data cleaning process was carried out to identify and address inconsistencies, missing values, and outliers. The reliability and validity of the instrument were supported by using previously validated tools and implementing inter-rater reliability checks where applicable. All collected data was securely stored to maintain confidentiality and data integrity throughout the research process.

## Data analysis

The data collected from the structured interview questionnaires were coded, entered, and analysed using STATA statistical software. Descriptive statistics such as frequencies, percentages, means, and standard deviations were used to summarise participants' sociodemographic characteristics and responses, providing an overall picture of patient satisfaction levels and the distribution of influencing factors.

The outcome variable, patient satisfaction, was categorised into "satisfied" and "unsatisfied" based on the mean satisfaction score calculated from 21 indicators. Patients scoring above the mean satisfaction score were classified as satisfied [20]. To identify factors associated with patient satisfaction, multivariable logistic regression analysis was performed. Model adequacy was evaluated using the Hosmer-Lemeshow goodness-of-fit test and assessments of multicollinearity among predictor variables. The Hosmer-Lemeshow test was applied to examine the calibration of the logistic regression model by comparing observed and expected outcomes across deciles of predicted risk. A non-significant p-value was considered indicative of acceptable model fit. Multicollinearity was assessed using variance inflation factors (VIFs), with values greater than 5 indicating moderate collinearity. Prior to final model selection, categorical predictors exhibiting moderate multicollinearity were recoded to ensure model stability and interpretability. The final model demonstrated satisfactory calibration and acceptable VIF values, confirming that the assumptions of logistic regression were met.

Results were presented using tables, charts, and graphs to enhance clarity and facilitate interpretation. A p-value of less than 0.05 was considered statistically significant to determine the strength and relevance of associations.

## Ethical considerations

Ethical approval for this study was obtained from the Institutional Review Board (IRB) of Yekatit 12 Medical College with reference number RPO/934/25. Prior to data collection, written informed consent was obtained from all participants after a clear explanation of the study's objectives, procedures, and their rights was provided. Participation was entirely voluntary, and participants were informed that they could withdraw from the study at any time without any consequences.

To maintain confidentiality, no personally identifying information was collected, and all data were securely stored and accessed only by the research team. The study adhered to established ethical principles, including respect for persons, beneficence (minimising harm), and justice (ensuring fairness and equity) in participant selection and treatment.

## Results

### Sociodemographic characteristics

A total of 422 respondents participated in the study. Of these, 59.0% were female. Regarding age distribution, the largest proportion of participants was in the 35–49 years category, accounting for 40.0% of the sample. Most respondents (93.13%) were from urban areas. In terms of religion, 64.93% identified as Orthodox Christians, followed by 19.67% who identified as Protestants. Regarding educational status, 42.65% had completed secondary education, while 31.99% had attained higher education. Concerning household income, more than half of the respondents (56.16%) reported a monthly income of ≤7168 ETB. Furthermore, many participants (92.89%) were residents of Addis Ababa. (Table 1).

### Patient experiences with pharmacy services

Among the total respondents, 67.77% were chronic care patients, while 32.23% were visiting for the first time. Regarding self-assessed health status, 70.38% considered themselves to be sick, while 29.62% rated their health as severely ill. In

Table 1. Sociodemographic characteristics of respondents (n = 422), Yekatit 12 Hospital, 2025.

| Variable | Category | Frequency | Percentage |
|---|---|---|---|
| Sex | Male | 173 | 41.00 |
| | Female | 249 | 59.00 |
| Age | 18-24 | 41 | 9.72 |
| | 25-34 | 120 | 28.44 |
| | 35-49 | 173 | 40.00 |
| | >50 | 88 | 20.85 |
| Place of resident | Rural | 29 | 6.87 |
| | Urban | 393 | 93.13 |
| Marital status | Single | 99 | 23.46 |
| | Married | 298 | 70.62 |
| | Divorce | 15 | 3.55 |
| | Widowed | 10 | 2.37 |
| Religion | Orthodox | 274 | 64.93 |
| | Muslim | 61 | 14.45 |
| | Protestant | 83 | 19.67 |
| | Other | 4 | 0.95 |
| Education status | No formal education | 27 | 6.40 |
| | Primary education | 80 | 18.96 |
| | Secondary and higher | 315 | 74.64 |
| Occupation | No | 25 | 5.92 |
| | Yes | 397 | 94.08 |
| Wealth | ≤7168 | 237 | 56.16 |
| | >7169 | 185 | 43.84 |
| Region | Addis Ababa | 392 | 92.89 |
| | Other | 30 | 7.11 |

terms of medication dispensed, 56.40% received either none or some of the prescribed medications. Regarding payment modality, 41.94% paid out of pocket, 35.78% had their costs covered by insurance, and 22.27% received care for free. Finally, 62.09% of participants experienced a waiting time of less than 23 minutes (Table 2).

### Satisfaction level

The satisfaction level of patients varied across different service areas within the hospital. The highest satisfaction was reported at the ART clinic, with 82.14% of respondents expressing satisfaction. This was followed by the Outpatient Department (OPD), with 57.65% of patients being satisfied. In the Inpatient Department (IPD), 50.59% of patients were satisfied. The Paediatrics ward recorded a lower satisfaction rate of 40.48%. The lowest satisfaction was observed in the Emergency Department, where only 17.86% of patients reported a satisfactory experience.

### Participant opinions towards the pharmacy setting, medication availability, and cost

Client satisfaction varied across the different components of pharmacy services. More than half of respondents were satisfied with the convenience of the pharmacy location (53.32%), although 25.12% reported dissatisfaction. Dissatisfaction was highest for the comfort of the private counselling area, where 57.58% of clients expressed dissatisfaction and only 25.83% were satisfied. Cleanliness received the most favourable ratings, with 62.56% satisfied with pharmacy cleanliness and 66.59% satisfied with dispensary cleanliness, while dissatisfaction in both areas remained below 11%. In contrast, the availability of needed medications showed a mixed response, with 42.65% satisfied but a substantial 38.86% dissatisfied, indicating notable concerns regarding medication stock and access (Fig 1)

### Participant satisfaction with the pharmacist's approach or communication

Most respondents expressed satisfaction with the pharmacy services provided. Specifically, 92.42% were satisfied with the availability of pharmacy professionals, while 83.65% were satisfied with the clarity of communication from pharmacy personnel. Around 82% of participants reported satisfaction with fairness in service provision, and 77.01% were satisfied with the politeness and interest demonstrated by pharmacy professionals. Additionally, 73.22% of respondents felt that pharmacists treated clients with respect. However, satisfaction with service waiting time was notably lower, with only 41.47% of clients reporting satisfaction, indicating that waiting time remains a significant concern for many patients (Fig 2).

**Table 2. Patient experiences with pharmacy services in Yekatit 12 Hospital, Addis Ababa.**

| Variable | Category | Frequency | Percentage |
|---|---|---|---|
| Familiarity with the institution | First visit | 136 | 32.23 |
| | Chronic care | 286 | 67.77 |
| Self-judge health status | Severely sick | 125 | 29.62 |
| | Sick | 297 | 70.38 |
| Medication dispensed | All | 184 | 43.60 |
| | None or some | 238 | 56.40 |
| Payment modality | Out-of-pocket | 177 | 41.94 |
| | Paid by insurance | 151 | 35.78 |
| | Free | 94 | 22.27 |
| Waiting time | <23min | 262 | 62.09 |
| | >23min | 160 | 37.91 |

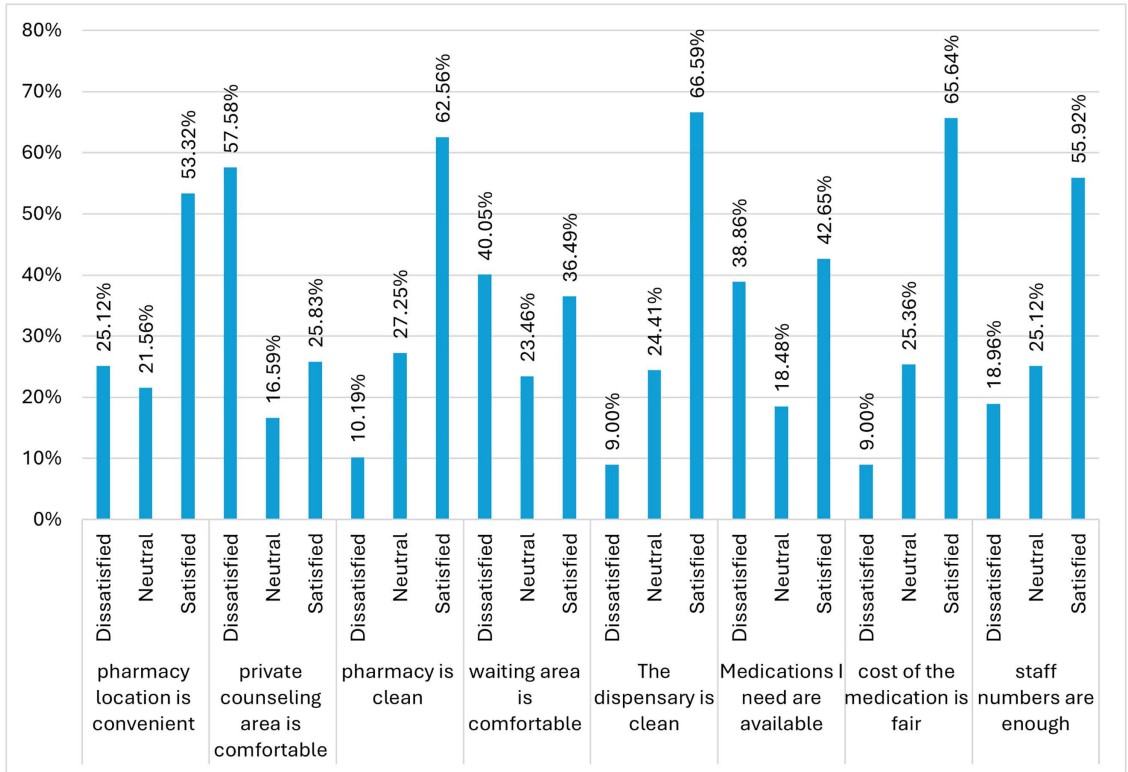

**Fig 1. Study participants' opinions towards pharmacy setting, medication availability, and cost.**

### Participant satisfaction towards the pharmacist's medication instructions

Client-reported satisfaction with pharmacy counselling and communication varied considerably across service components. Nearly half of respondents (46.21%) were dissatisfied with the time allocated for counselling, making it one of the poorest-rated elements. In contrast, communication clarity was rated highly, with 92.65% satisfied with receiving instructions in an understandable language. Similarly, most participants (79.86%) were satisfied with understanding how to take their medications as prescribed. However, the majority (70.62%) of respondents were dissatisfied with the explanation of the medication's precautions and side effects. In addition, guidance on proper medication storage was a major weakness, with more than half (51.66%) being dissatisfied (Fig 3).

### Study participants' satisfaction score

The satisfaction score ranges from 41 to 98. The mean score was 69.92 with a standard deviation of ±8.85, which was used as a cut point to categorise participants as satisfied and not satisfied. The overall client satisfaction with pharmacy services was 49.76%, while 50.24% of clients were dissatisfied (Table 3).

### Determinants of patients' level of satisfaction and associated factors with pharmacy service

In the bivariate analysis using crude odds ratios (COR), several factors were found to be significantly associated with client satisfaction toward pharmacy services. Urban residents were less likely to be satisfied compared to rural residents (COR = 0.42, 95% CI: 0.19,0.95). Clients with secondary education (COR = 0.26, 95% CI: 0.11,0.62) were also significantly less likely to be satisfied compared to those with no formal education. Clients who did not receive all their prescribed

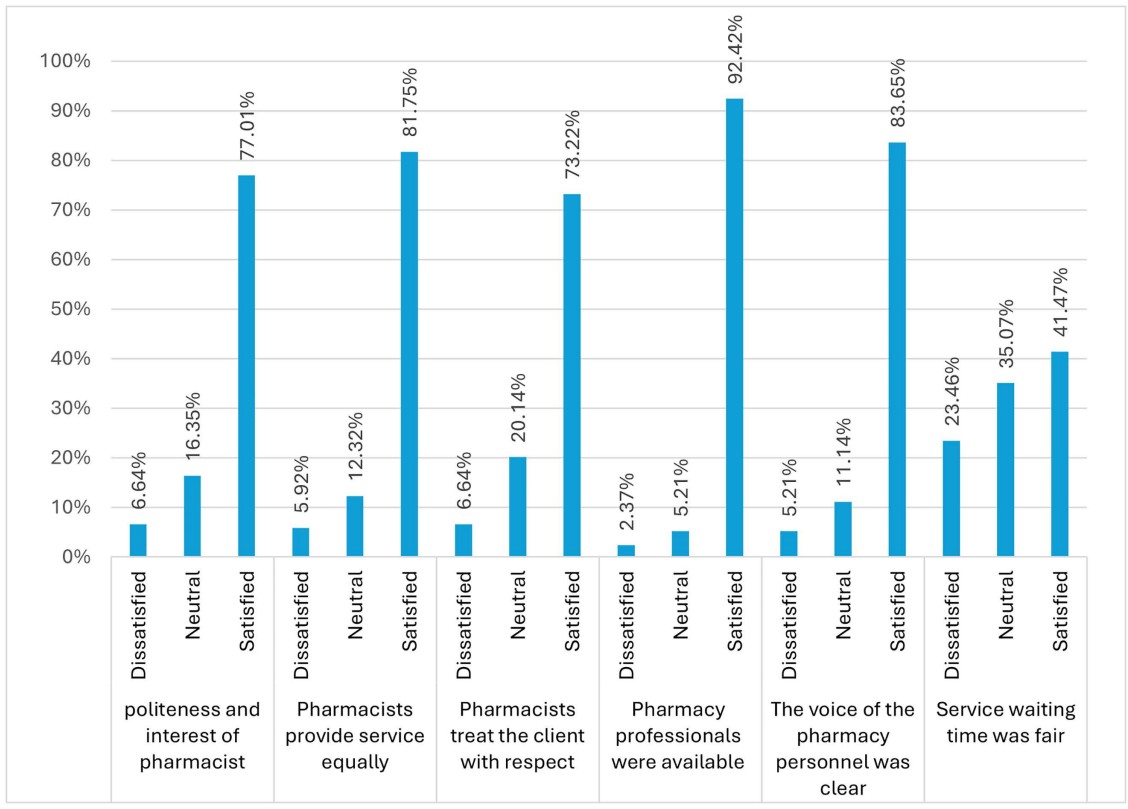

**Fig 2. Study participants' satisfaction towards the pharmacist's approach or communication.**

medications showed lower satisfaction (COR = 0.29, 95% CI: 0.19, 0.43) compared to those who received all their medicines. In terms of income, participants with higher incomes (> 7,169 ETB) had lower satisfaction levels (COR = 0.54, 95% CI: 0.36, 0.79). Pharmacy type was also associated with satisfaction. Clients attending the paediatrics pharmacy (COR = 0.50, 95% CI: 0.27–0.92) and the emergency pharmacy (COR = 0.16, 95% CI: 0.08,0.32) were less likely to be satisfied, while those attending the ART pharmacy were more likely to be satisfied (COR = 3.38, 95% CI: 1.67–6.84), compared to those using the outpatient pharmacy.

In the multivariable logistic regression analysis using Adjusted Odds Ratio (AOR), clients with secondary and above education (AOR = 0.24, 95% CI: 0.07,0.80) were less likely to be satisfied compared to those with no formal education. Not receiving all prescribed medications remained a strong predictor of dissatisfaction (AOR = 0.35, 95% CI: 0.20,0.60). Additionally, clients who received services from the ART pharmacy (AOR = 3.28, 95% CI: 1.38, 7.78) were significantly more likely to be satisfied compared to those who used the outpatient pharmacy (Table 4).

## Discussion

This study assessed client satisfaction and the associated factors with pharmacy services across various service outlets at Yekatit 12 Hospital, specifically OPD, IPD, paediatrics, ART, and emergency pharmacies. In multivariate analyses, some socio-demographic and service-related factors, such as Education, medication dispensed, and pharmacy type, were significantly associated with patient satisfaction.

Overall, 49.76% of the participants were satisfied with the outpatient pharmaceutical services. It is also slightly higher than findings from Yekatit 12 Referral Hospital's previous assessment, which reported a satisfaction level of 47% [20].

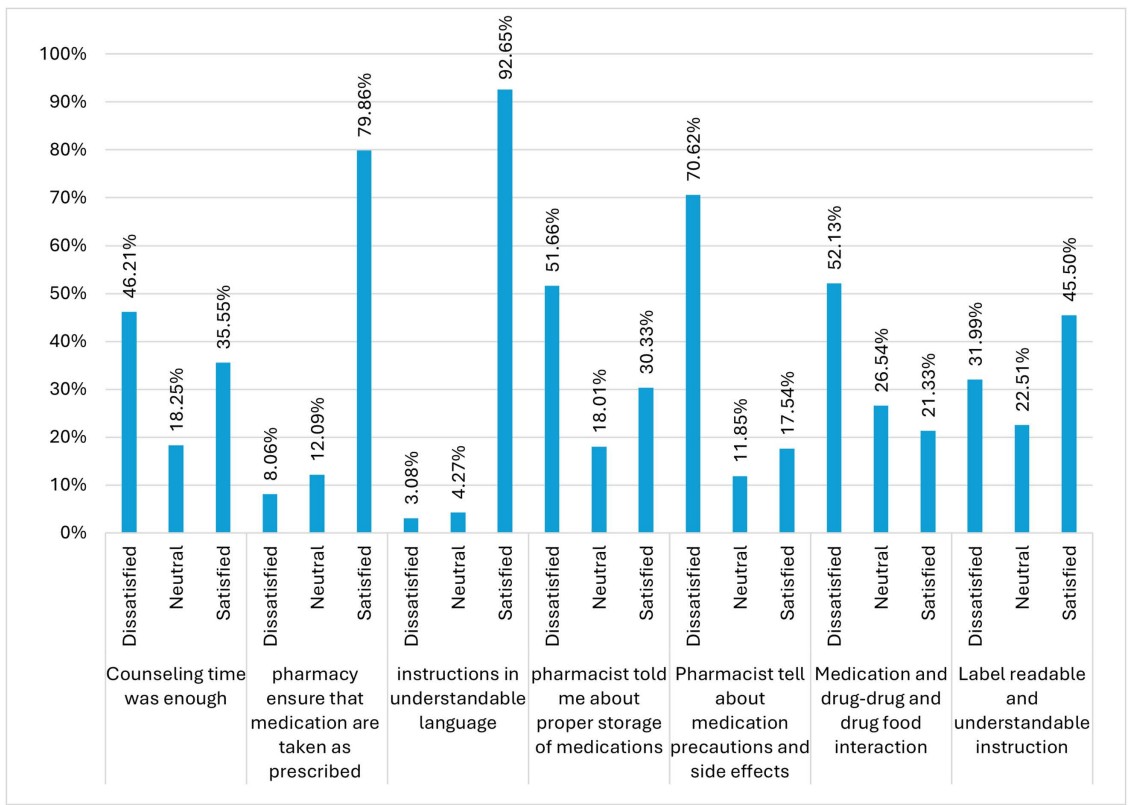

**Fig 3. Study participants' satisfaction with the pharmacist's medication instructions.**

**Table 3. Study participants' satisfaction score towards pharmacist services.**

| Summary statistics | Frequency | Percentage |
|---|---|---|
| Mean score (St. deviation) | 69.92 ± 8.85 | |
| Median score (range) | 69(41-98) | |
| Mode | 66 | |
| Satisfaction score >69.9 (satisfied) | 206 | 49.76% |
| Satisfaction score ≤ 69.9 (dissatisfied) | 209 | 50.24% |

Furthermore, the result is in line with studies conducted at Wolaita Sodo Teaching Hospital (54.2%) [21] and Jimma University Specialised Hospital (57.1%) [22]. Although the satisfaction score in our study was generally lower. The higher mean score is attributed to improved service quality, more organised pharmacy workflow, and enhanced communication strategies in specialty pharmacies such as ART and paediatrics.

Compared to high-income countries like South Korea (74.6%) and the United Arab Emirates (77.1%), the satisfaction level in Yekatit 12 Hospital was found to be low [23,24]. The higher satisfaction rates observed in these studies may reflect the presence of more developed and better-resourced pharmacy service systems in high-income countries compared to those in developing countries, such as Ethiopia [25].

**Table 4. Bivariable and Multivariable Logistic Regression Analysis of Factors Associated with Patients' Satisfaction.**

| Variable | Category | Satisfaction | | Odds ratio | |
|---|---|---|---|---|---|
| | | Dissatisfied | Satisfied | COR (95% CI) | AOR (95% CI) |
| Age | 18-24(Ref) | 20 (48.78) | 21 (51.22) | | |
| | 25-34 | 69 (57.50) | 51 (42.50) | 0.70 (0.35,1.43) | 0.94 (0.38, 2.33) |
| | 35-49 | 89 (51.45) | 84 (48.55) | 0.90 (0.45,1.78) | 0.82 (0.32, 2.13) |
| | >50 | 34 (38.64) | 54 (61.36) | 1.51 (0.72,3.19) | 1.06 (0.37, 3.10) |
| Sex | Male (Ref) | 88 (50.87) | 85 (49.13) | | |
| | Female | 124 (49.80) | 125 (50.20) | 1.04 (0.71,1.54) | 0.81 (0.49, 1.36) |
| Marital status | Single (Ref) | 51 (51.52) | 48 (48.48) | | |
| | Married | 147 (49.33) | 151 (50.67) | 1.09 (0.69,1.72) | 1.51 (0.74, 3.05) |
| | Divorced | 8 (53.33) | 7 (46.67) | 0.93 (0.31,2.76) | 0.88 (0.15, 5.03) |
| | Widowed | 6 (60.00) | 4 (40.00) | 0.71 (0.19,2.67) | 0.16 (0.02, 1.09) |
| Resident | Rural (Ref) | 9 (31.03) | 20 (68.97) | | |
| | Urban | 203 (51.63) | 190 (48.35) | 0.42 (0.19,0.95)* | 0.31 (0.08, 1.20) |
| Region | Amhara (Ref) | 199 (50.77) | 193 (49.23) | | |
| | Other regions | 13 (43.33) | 17 (56.67) | 1.35 (0.64,2.85) | 0.56 (0.15, 2.15) |
| Religion | Orthodox (Ref) | 133 (48.54) | 141 (51.46) | | |
| | Muslim | 30 (49.18) | 31 (50.82) | 0.97 (0.56,1.70) | 0.88 (0.44, 1.74) |
| | Protestant | 47 (56.63) | 36 (43.37) | 0.72 (0.44,1.18) | 0.97 (0.52, 1.82) |
| | Other | 2 (50.00) | 2 (50) | 0.94 (0.13,6.79) | 0.77 (0.05, 12.25) |
| Education | No education (Ref) | 7 (25.93) | 20 (74.07) | | |
| | Primary education | 23 (28.75) | 57 (71.25) | 0.87 (0.32,2.33) | 0.59 (0.17, 2.10) |
| | Secondary and higher education | 182 (57.78) | 133 (42.22) | 0.26 (0.11,0.62)* | 0.24 (0.07, 0.80)* |
| Medication dispensed | All (Ref) | 61 (33.15) | 123 (66.85) | | |
| | None or some | 151 (63.45) | 87 (36.55) | 0.29 (0.19,0.43)* | 0.35 (0.20, 0.60)* |
| Occupation | No (Ref) | 8 (32.00) | 17 (68.00) | | |
| | Yes | 204 (51.39) | 193 (48.61) | 0.45 (0.19,1.06) | 0.69 (0.25, 1.90) |
| Income | ≤ 7168 (Ref) | 103 (43.46) | 134 (56.54) | | |
| | >7169 | 109 (58.92) | 76 (41.08) | 0.54 (0.36,0.79)* | 0.85 (0.49, 1.45) |
| Waiting time | ≤23 min (Ref) | 130 (49.62) | 132 (50.38) | | |
| | >23 min | 82 (51.25) | 78 (48.75) | 0.94 (0.63,1.39) | 0.23 (0.12, 0.46) * |
| Pharmacy type | Outpatient pharmacy (Ref) | 36 (42.35) | 49 (57.65) | | |
| | Inpatient pharmacy | 42 (49.41) | 43(50.59) | 0.75 (0.41,1.38) | 0.42 (0.19, 0.90) * |
| | Paediatrics pharmacy | 50 (59.52) | 34 (40.48) | 0.50 (0.27,0.92)* | 0.24 (0.11, 0.56) * |
| | Antiretroviral pharmacy (ART) | 15 (17.86) | 69 (82.14) | 3.38 (1.67,6.84)* | 3.28 (1.38, 7.78) * |
| | Emergency pharmacy | 69 (82.14) | 15 (17.86) | 0.16 (0.08,0.32)* | 0.09 (0.04, 0.22) * |

*Indicates a p-value less than 0.05.*

Medication availability emerged as a key determinant of patient satisfaction in this study. This finding is consistent with previous studies conducted in Ethiopian healthcare settings, which have identified the consistent availability of medications as a major contributor to positive patient experiences [25,26]. In contrast, the unavailability of prescribed medications may compel patients to seek alternatives from external sources, often resulting in additional financial burden and decreased satisfaction with the healthcare service [11]. In the present study, only 43.86% of respondents reported receiving all their prescribed medications from the hospital pharmacy. Patients who were able to obtain the full set of prescribed

drugs demonstrated significantly higher satisfaction levels compared to those who received only some or none of their medications [27,28]. This association was statistically significant, with an adjusted odds ratio (AOR) of 0.37, indicating that incomplete medication dispensing is a strong predictor of reduced patient satisfaction.

The type of pharmacy service patients accessed significantly influenced their satisfaction levels. Antiretroviral Therapy (ART) pharmacy users reported higher satisfaction compared to those in general outpatient pharmacies. In a study conducted in Dembia District, Ethiopia, patients attending ART pharmacy services exhibited a satisfaction rate of 75% [29]. This might be a result of the ART services having shorter waiting times, stronger provider-patient engagement, and tailored counselling, which may increase patient satisfaction [30,31]. On the other hand, emergency pharmacy users were less satisfied. This was consistent with a study conducted in Hawassa [32]. The possible reason for this might be low availability of medication due to the stock status of the pharmaceutical fund and supply agencies, and low budget allocation to the hospital pharmacy [33].

## Strengths and limitations of this study

One of the strengths of this study is its inclusion of various pharmacy service areas such as outpatient, inpatient, ART, paediatric, and emergency units, which provided a comprehensive overview of client satisfaction across different departments within the hospital. The relatively large sample size and use of a standardised, structured questionnaire also enhanced the reliability and consistency of the data collected. However, the study has several limitations. Being a cross-sectional design, it limits the ability to draw causal inferences between satisfaction and associated factors. Additionally, since the study was conducted at a single hospital, the findings may not be generalizable to other settings. There is also a potential for interviewer bias or social desirability bias, as interviews were conducted within the hospital premises. Moreover, the study did not assess other potentially influential factors, such as prior experiences with pharmacy services or patient expectations, which may have impacted satisfaction responses. Additionally, as the study relies on self-reported satisfaction may limit objectivity, as no triangulating data were collected.

## Conclusions

The overall client satisfaction with pharmacy services was was low, with less than half of the clients being satisfied with the outpatient pharmaceutical services. Among the various factors examined, place of residence, education level, type of pharmacy visited, and whether all prescribed medications were dispensed were significantly associated with client satisfaction. Notably, clients who accessed services in specialised pharmacies, such as the ART unit, and those who received all their medications reported higher satisfaction. These findings highlight the need for targeted improvements in drug availability, staff communication, and service delivery, particularly in general outpatient settings.

## Supporting information

**S1 File. Questionnaire.**
(DOCX)

## Author contributions

**Conceptualization:** Mekdes Tamiru Yizengaw, Bekele Simegn Demissie, Sosina Fikadu Mulugeta, Samrawit Birhanu Alemu.

**Data curation:** Mekdes Tamiru Yizengaw, Beimnet Taye Mekonnen, Sosina Fikadu Mulugeta, Ekram kemale Mohmmed, Samrawit Birhanu Alemu.

**Formal analysis:** Mekdes Tamiru Yizengaw, Bekele Simegn Demissie, Beimnet Taye Mekonnen, Birknesh Tamire Haile, Sosina Fikadu Mulugeta, Samrawit Birhanu Alemu, Melaku Birhanu Alemu.

**Investigation:** Mekdes Tamiru Yizengaw, Birknesh Tamire Haile, Melaku Birhanu Alemu.

**Methodology:** Mekdes Tamiru Yizengaw, Bekele Simegn Demissie, Beimnet Taye Mekonnen, Birknesh Tamire Haile, Sosina Fikadu Mulugeta, Samrawit Birhanu Alemu, Melaku Birhanu Alemu.

**Project administration:** Mekdes Tamiru Yizengaw, Birknesh Tamire Haile.

**Software:** Mekdes Tamiru Yizengaw, Ekram kemale Mohmmed.

**Supervision:** Bekele Simegn Demissie, Melaku Birhanu Alemu.

**Validation:** Mekdes Tamiru Yizengaw, Beimnet Taye Mekonnen.

**Visualization:** Mekdes Tamiru Yizengaw, Birknesh Tamire Haile, Melaku Birhanu Alemu.

**Writing – original draft:** Mekdes Tamiru Yizengaw.

**Writing – review & editing:** Mekdes Tamiru Yizengaw, Bekele Simegn Demissie, Beimnet Taye Mekonnen, Sosina Fikadu Mulugeta, Ekram kemale Mohmmed, Melaku Birhanu Alemu.

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
