## [Decision Letter · Decision Letter 0]

6 Oct 2025

PONE-D-25-38720Assessment of Client Satisfaction with Pharmacy Services and Associated Factors at Yekatit 12 Hospital Medical College, Addis Ababa, Ethiopia 2025PLOS ONE

Dear Dr. Yizengaw,

Thank you for submitting your manuscript to PLOS ONE. After careful consideration, we feel that it has merit but does not fully meet PLOS ONE’s publication criteria as it currently stands. Therefore, we invite you to submit a revised version of the manuscript that addresses the points raised during the review process.

We look forward to receiving your revised manuscript.

Kind regards,

Muhammad Atif

Academic Editor

PLOS ONE

Journal Requirements:

Reviewers' comments:

Reviewer's Responses to Questions

**Comments to the Author**

1. Is the manuscript technically sound, and do the data support the conclusions?

Reviewer #1: Yes

Reviewer #2: Partly

2. Has the statistical analysis been performed appropriately and rigorously? 

Reviewer #1: Yes

Reviewer #2: No

3. Have the authors made all data underlying the findings in their manuscript fully available?

Reviewer #1: Yes

Reviewer #2: Yes

4. Is the manuscript presented in an intelligible fashion and written in standard English?

Reviewer #1: No

Reviewer #2: No

5. Review Comments to the Author

Reviewer #1: Reviewer comments

This study highlighted the significance of assessing patient or customer satisfaction with pharmacy services as a key indicator of healthcare quality. It provided insights into how pharmacy practices impact patient trust, adherence and overall healthcare outcomes. However, I have comments that might improve the quality of manuscript.

1. “Client satisfaction” is understandable, but in academic and professional writing, alternatives such as patient or customer satisfaction are considered more precise and widely accepted.

2. Who were your clients? Were these patients or their caregivers?

3. Please mention the ethical approval offering body in abstract if you are writing that approval was obtained.

4. Grammatical mistakes are present throughout the manuscript, revise it for language check.

5. Can you please write some detail on your healthcare system? Pharmacists were clinical pharmacist or hospital pharmacist? What was their job description hospitals?

6. Can you provide the source of information regarding the hospital that you mentioned in your study setting to improve the reliability of data?

7. In methodology section, you mentioned that all patients were taken as clients. Should not you also include caregivers that generally approach pharmacies for medicines for in-patients? Or explain a bit about your hospital system in which how in-patients or bed-ridden patients can visit pharmacy?

8. If you were just dealing with the patients, should not it be patient satisfaction instead of client satisfaction?

9. In exclusion criteria, you have mentioned that clients below 18 years were excluded, while in abstract and results you have written that client who visited pediatric pharmacy, were significantly unsatisfied. It’s a bit confusing. Pediatric patients are above 18 years of age? How you have taken data from pediatric patients (below 18) who visited pediatric pharmacy?

10. What if any patient refused to response, how randomization was maintained?

11. For quantitative studies, generally structured tools are used for data collection. Why you used semi-structured questionnaire?

12. How you validated the study tool?

13. You did not explain about the scoring assumptions of your tool.

14. Discussion can be made rich while not only discussing your findings, but also providing global context.

15. In study strengths you are stating that “the relatively large sample size and use of a standardised, structured questionnaire also enhanced the reliability and consistency of the data collected: while above you have mentioned that you used semi-structured tool. It is confusing.

Reviewer #2: Please refer to my attached report for the detailed reviews. In that report, I have provided comprehensive comments addressing the strengths, weaknesses, and areas for improvement in the manuscript. These comments should serve as the basis for further revision and consideration.

6. PLOS authors have the option to publish the peer review history of their article (what does this mean?). If published, this will include your full peer review and any attached files.

Reviewer #1: No

Reviewer #2: **Yes:** Fahad Saleem

---

## [Author Response · Author response to Decision Letter 1]

9 Dec 2025

We have attached a point-by-point response to the reviewer's comment.

---

## [Decision Letter · Decision Letter 1]

1 Apr 2026

PONE-D-25-38720R1Client satisfaction with pharmacy services and associated factors at Yekatit 12 Hospital Medical College, Addis Ababa, Ethiopia 2025PLOS One

Dear Dr. Yizengaw,

Thank you for submitting your manuscript to PLOS ONE. After careful consideration, we feel that it has merit but does not fully meet PLOS ONE’s publication criteria as it currently stands. Therefore, we invite you to submit a revised version of the manuscript that addresses the points raised during the review process.

A letter that responds to each point raised by the academic editor and reviewer(s). You should upload this letter as a separate file labeled 'Response to Reviewers'.

We look forward to receiving your revised manuscript.

Kind regards,

Helen Howard

Staff Editor

PLOS One

Journal Requirements:

Additional Editor Comments:

- Please ensure that the references are formatted using the Vancouver style, as per the PLOS ONE submission guidelines https://journals.plos.org/plosone/s/submission-guidelines. In particular, please ensure that each citation in the reference list states the journal name.

Reviewers' comments:

Reviewer's Responses to Questions

**Comments to the Author**

1. If the authors have adequately addressed your comments raised in a previous round of review and you feel that this manuscript is now acceptable for publication, you may indicate that here to bypass the “Comments to the Author” section, enter your conflict of interest statement in the “Confidential to Editor” section, and submit your "Accept" recommendation.

Reviewer #2: All comments have been addressed

2. Is the manuscript technically sound, and do the data support the conclusions?

Reviewer #2: Yes

3. Has the statistical analysis been performed appropriately and rigorously? 

Reviewer #2: Yes

4. Have the authors made all data underlying the findings in their manuscript fully available?

Reviewer #2: Yes

5. Is the manuscript presented in an intelligible fashion and written in standard English?

Reviewer #2: Yes

6. Review Comments to the Author

Reviewer #2: All comments are addressed. The authors have conducted a through revision. The manuscript looks good in the current form.

7. PLOS authors have the option to publish the peer review history of their article (what does this mean?). If published, this will include your full peer review and any attached files.

Reviewer #2: **Yes:** Fahad Saleem

---

## [Author Response · Author response to Decision Letter 2]

14 Apr 2026

There were no reviewer comments. However, we have provided a response to the editor’s comment.

---

## [Editor Report · Decision Letter 2]

28 Apr 2026

Client satisfaction with pharmacy services and associated factors at Yekatit 12 Hospital Medical College, Addis Ababa, Ethiopia

PONE-D-25-38720R2

Dear Dr. Yizengaw,

We’re pleased to inform you that your manuscript has been judged scientifically suitable for publication and will be formally accepted for publication once it meets all outstanding technical requirements.

Kind regards,

Helen Howard

Staff Editor

PLOS One
---

## [Editor Report · Acceptance letter]

PONE-D-25-38720R2

PLOS One

Dear Dr. Yizengaw,

I'm pleased to inform you that your manuscript has been deemed suitable for publication in PLOS One. Congratulations! Your manuscript is now being handed over to our production team.

Kind regards,

on behalf of

Dr Helen Howard

Staff Editor

PLOS One